# Leveraging Nutrient-Rich Traditional Foods to Improve Diets among Indigenous Populations in India: Value Chain Analysis of Finger Millet and Kionaar Leaves

**DOI:** 10.3390/foods11233774

**Published:** 2022-11-23

**Authors:** Shauna M. Downs, Ridhima Kapoor, Emily V. Merchant, Tamara Sullivan, Geetanjali Singh, Jessica Fanzo, Suparna Ghosh-Jerath

**Affiliations:** 1Department of Health Behavior, Society and Policy, Rutgers School of Public Health, Newark, NJ 07102, USA; 2The George Institute for Global Health India, New Delhi 110025, India; 3New Use Agriculture and Natural Plant Products Program, Department of Plant Biology, Center for Agricultural Food Ecosystems, The New Jersey Institute for Food, Nutrition and Health, Rutgers University, New Brunswick, NJ 08901, USA; 4Department of Urban-Global Public Health, Rutgers School of Public Health, Newark, NJ 07102, USA; 5Department of Botany, Dr. Shyama Prasad Mukherjee University Ranchi, Ranchi 834008, India; 6Berman Institute of Bioethics, Nitze School of Advanced International Studies (SAIS), Bloomberg School of Public Health, Johns Hopkins University, Washington, DC 20036, USA

**Keywords:** finger millet, Indigenous foods, Koinaar leaves, nutrient-rich foods, traditional ecological knowledge, value chain analysis

## Abstract

Many indigenous foods are nutrient-rich but are often underutilized even among populations at high risk of malnutrition. The aims of this study were to conduct value chain analysis of one cultivated crop (finger millet among the Munda tribe) and one wild green leafy vegetable (Koinaar leaves among the Sauria Paharia tribe) of two Indigenous communities in Jharkhand state, India and to identify entry points for interventions aimed at supporting production and consumption. Semi-structured interviews were conducted with stakeholders among each tribal group and transcripts were open coded and organized based on key themes across the steps of the value chain for each food independently. Improved storage techniques and infrastructure, machinery for processing and improved cooking fuel would help reduce barriers across the finger millet supply chain related to postharvest losses, processing labor and safety concerns related to cooking. For Koinaar leaves, improving drying techniques to increase consumption across seasons and providing training and support to increase opportunities for selling leaves in local markets, where participants mentioned potential language barriers, could strengthen the supply chain. Improving extension services and focusing beyond production has potential to improve the production and consumption of both nutrient-rich crops among Indigenous communities in India.

## 1. Introduction

Indigenous food systems, defined as food systems that rely on natural flora and fauna and have emerged through adaptation between people and place [1], have the potential to be leveraged to improve dietary diversity, nutrition and health outcomes [2]. India is home to 705 Indigenous communities who live in rural areas, several of them residing amidst a rich biodiversity, giving them access to diverse plant flora including wild edible plants and traditional crops [3,4,5,6]. Knowledge about the production, preservation and consumption of these plants is engrained into the existing traditional ecological knowledge (TEK) of these communities; however, they have been underutilized in helping to address malnutrition [2].

Indigenous communities in India procure their food from both natural and built food environments [7]. The foods sourced from natural environments comprise cultivated and wild plant foods and livestock that are produced, collected and raised for household consumption as well as for sales. A rich diversity of wild edible plants is generally collected from diverse habitats such as forests, cultivable fields, homesteads, roadsides and wastelands [5]. These wild edible plants are known to provide important nutrients such as beta-carotene, folic acid, vitamin C, calcium, iron and zinc and thus could be a local and affordable solution to address malnutrition in India [3,5]. In addition to wild edible plants, Indigenous populations have a long history of cultivating nutrient-rich varieties of staple crops [8]. While there has been a transition away from the cultivation of these crops, many of these traditional crops may have properties that make them more resilient to climate shocks [9] in addition to being nutrient-rich. The decline in consumption of traditional foods may also have implications on the micronutrient status of the vulnerable Indigenous population. For example, there has been a decreasing trend in consumption of coarse cereals with an associated decline in iron intake among the Indian population, especially in states where rice has replaced coarse cereals as the staple food [10].

A value chain analysis identifies places along the supply chain where value can be added to the final product by examining the “full range of activities that are required to bring a food product from conception, through the different phases of production, to delivery to final consumers and disposal after use” [11]. These activities as well as their corresponding facilitators and barriers affect the value of the final product [11]. Nutrition-focused value chain analysis allows for the identification of nutrition entry and exit points, maximizing nutritional value, across the steps of the food supply chain. Entry points are places where nutrition can be enhanced along the supply chain such as through fortification, improvements in processing and behavior change communication [11,12], whereas exit points are places where nutrition is lost due to inefficient harvest, lack of storage, ultra-processing, etc. [11,12]. In addition, identifying entry and exit points for nutrition across the supply chain also allows for the identification of interventions aimed at addressing them.

There is existing literature examining the value chains of crops in India, such as pearl and finger millet [13,14,15]; however, this research tends to focus on longer, industrial value chains with policy recommendations targeting larger stakeholder groups and activities such as market aggregation and distribution. To capture traditional ecological knowledge for improved human and planetary health, it is critical that analyses of short value chains are conducted at a more local level for wild edible plants, such as the Koinaar tree and traditionally consumed staple crops, such as finger millet. The current literature highlights the nutritional and ecological contributions of wild edible foods and traditional staple crops within Indigenous communities in India [2,3,5,6,8]; however, little is known about how their value chains can be leveraged to improve their production and consumption.

There is a clear need to conduct value chain analyses of underutilized crops within Indigenous communities in India given that this can help to identify how supply chains can be strengthened to improve their production, the way they move through the supply chain, as well as their consumption and contribution towards nutrient intake, thus adding nutritional value to the supply chain. The overarching research question for this study was: How can the value chains of the Koinaar tree and finger millet be leveraged to improve their production and consumption among Indigenous communities in India? The specific aims were to conduct value chain analysis of one cultivated (finger millet) and one wild (Koinaar tree) nutrient-rich and climate-resilient crop with two Indigenous communities in Jharkhand state, India, and to identify entry points for interventions or policies aimed at supporting their added value from production to consumption.

## 2. Methods

### 2.1. Overview of Methods

The present study was exploratory in nature and was part of a larger study which aims to understand the contribution of indigenous foods to nutrient intake and dietary diversity among the indigenous women and children of Jharkhand state, India [16]. As part of this larger study, we conducted two value chain analysis case studies of nutrient-rich foods among Indigenous tribes in Jharkhand State. More specifically, we conducted an analysis of finger millet (local name: mandua) within the Munda community and the leaves of the Koinaar tree (*Bauhinia purpurea* L.) within the Sauria Paharia community in Jharkhand. In each of the communities, we conducted semi-structured interviews to gain insight into the different steps of the value chain and into barriers and facilitators for the production, acquisition and consumption of the plants.

### 2.2. Rationale for Conducting Value Chain Analysis of Finger Millet and Koinaar Tree Leaves

Finger millet is a cereal that can serve as a source of vitamins and minerals but is not currently being fully utilized among the Munda tribal communities. In particular, it has high quantities of several nutrients including thiamine, calcium and dietary fiber [14]. In our previous work, finger millet was one of many Indigenous foods mentioned by participants, but it was infrequently consumed due to its low agricultural production, despite it having a favorable taste [17]. One of the main reasons cited by the community for declining cultivation of finger millet was the shift towards cultivation of hybrid rice crop as this provides better yield which is associated with higher monetary gains [17].

The leaves of the Koinaar tree (local name: Komo Ghasi) were one of several Indigenous foods reported to be consumed in previous work conducted with the Sauria Paharia community [2]. Koinaar leaves are an Indigenous green leafy vegetable, which has been found to be a rich source of several important nutrients including vitamin A, thiamine, calcium and iron, and thus may have potential to improve nutritional outcomes [2]. Within the Sauria Paharia community, Koinaar leaves are known for their “flavorsome taste and satiety giving properties” [2]. It is a small to medium-sized deciduous evergreen tree attaining a height of about 15 to 20 feet. The young tender leaves, flowers and floral buds are the edible parts of the Koinaar tree. The tender, delicate, new leaves can be observed on this deciduous tree species during early summer in the months from March to May, whereas the flowers and floral buds can be seen after rainy season in the months from October to November.

### 2.3. Overview of Study Context

This study is part of a series of studies focused on Jharkhand, a state in central eastern India [16]. Jharkhand was purposively selected because it is one of several states characterized as poorly performing on nutrition indicators such as chronic energy deficiency in women, high rates of stunting, wasting and underweight in children and high prevalence of maternal and childhood anemia, and as such faces a high burden of malnutrition [18]. It is also the state with the highest proportion of Indigenous communities in India. Within Jharkhand, there is a large Indigenous population (26.2%), who in India are referred to as Scheduled Tribes (ST) based on their traditional traits, distinctive culture, geographical isolation and poor socioeconomic status, with nearly static levels of progress [19]. In previous work, we identified four of these tribes to focus on, the Santhals, Ho, Munda and Sauria Paharia [2,16,17]. The first three of these tribes were chosen because of their high population in the state [16]. The Sauria Paharia tribe is smaller, making up less than 1% of the Scheduled Tribe population [16]. However, they have been identified as particularly vulnerable due to their low literacy levels, pre-agricultural system of existence and zero or negative population growth [4,16]. The two case studies included in this manuscript are focused on the Munda and Sauria Paharia tribal groups.

The Mundas account for 14.8% of the total state tribal population in Jharkhand and mainly inhabit the Chotanagpur region, which comprises the Khunti district [20,21]. The Mundas are mainly dependent upon subsistence agriculture; however, they also engage in foraging, hunting and livestock breeding to supplement their economy [22]. Within the Khunti district, two blocks (Torpa and Murhu) were identified for participation with further selection of two villages per block (Torpa: Tati and Nichitpur villages; Murhu: Charid and Kudapurti villages) (Figure 1).

The Sauria Paharia tribe is a particularly vulnerable tribal group that resides in the hilly, dense forest regions of Jharkhand [23,24]. The study was conducted in the Godda district of Jharkhand, which is home to the Sauria Paharia community (total population: 13,688 people) [4]. The district lies in the northeastern part of Jharkhand, with a total geographical area of 2110.4 km^2^ and is surrounded by hills and small forests, comprising 11.2% of the total area [25]. Two geographically diverse blocks, Sunderpahari and Boarijor in the Godda district of Jharkhand (Figure 1), were purposively selected based on their high population of Sauria Paharia tribal members. Four villages (Sunderpahari: Chewo and Chota Haripur villages; Boarijor: Kusumghati and Teletok villages) were included in the data collection in order to account for variation in food access. We provide additional information about each of these communities in their respective case studies.

### 2.4. Semi-Structured Interviews for Value Chain Analysis

Semi-structured interviews focusing on the steps of the food supply chain were conducted in person among the Mundas (January–April 2022) and Sauria Paharia (November 2020–February 2021) communities. The interview guides focused on the different steps of the value chain and the barriers and facilitators that influenced the way the plant moved along the chain. The interview guides were tailored to each plant and took approximately 15–30 min to complete. Interviews among each tribal community were conducted with the following stakeholder groups: (1) village leaders, (2) healthcare and nutrition workers and (3) village members (selected using snowball sampling technique). A total of 12 interviews were conducted as part of the finger millet value chain analysis in the Munda tribe (see Appendix A for description of participant characteristics). Among the Munda tribe, community members with some experience with farming finger millet (either past or present) were purposively selected to participate in the interviews among these stakeholder groups. A total of 19 interviews were conducted as part of the Koinaar tree value chain analysis in the Sauria Paharia community (Appendix A). Among the Sauria Paharia community, female respondents were purposively selected for the interviews, as they usually engaged in food collection and preparation activities and thus, were expected to provide more information on the key themes. All the interviews were conducted in local languages by trained local field investigators from the respective communities. The interview recordings were transcribed from the local languages to Hindi and subsequently translated to English for data analysis. The schematic workflow of the study is presented in Figure 2.

### 2.5. Data Analysis

Interview transcripts were open coded and organized based on key themes related to the barriers and facilitators at each step of the value chain using NVivo (QSR International Pty Ltd., Doncaster, Australia, Version 12) software. Following coding and analysis, a value chain was constructed to highlight the barriers and facilitators at each point along the value chain. We then identified interventions across the supply chains that could help to address these barriers, or leverage the facilitators, to increase the production and consumption of each plant.

We received ethical approval from the Institutional Ethics Committee at the Indian Institute of Public Health-Delhi, Public Health Foundation of India and All India Institute of Medical Sciences, New Delhi. Administrative approvals from authorities at the district level as well as cluster level consent from the village leader were also obtained. Literate respondents provided written informed consent while third-party witnessed verbal consents were taken from those who could not read or write.

## 3. Case Study 1: Finger Millet Value Chain among Munda Tribe

### 3.1. Production

Figure 3 provides an overview of the steps of the finger millet’s supply chain. Participants described cultivating finger millet since “ancient time” and noted that they continue to do so today for food and revenue: “We cultivate (mandua) and we use it in our meal… When there is some need at home then we sell it…” (Female health and nutrition worker, age 35, Charid village, Murhu block). However, several participants noted that they are not currently growing mandua, with some noting the production of rice in replacement: “Now we are not doing because apart from rice we don’t grow anything” (Male village head, age 54, Tati village, Torpa block).

A few inputs were described as being needed for its production. Farmers used desi or local seeds and reported saving them from one year to the next. Just over half of participants reported using organic or inorganic fertilizer and a few reported using machinery (i.e., tractor) during cultivation. In terms of production, participants reported travelling between 10 to 35 min to reach the “Goda” farm where they produced finger millet. It is of note that the Munda tribe cultivates crops at three levels of farmlands, namely Loyong (low level lands with highest water requirement for crops), Badi (middle level lands with relatively low water requirement for crops) and Goda (dry stony plain lands with least water requirement). Cultivation was typically described as taking place in July with harvesting in October–November. However, some participants described delays to crop plantation due to insufficient labor: “Sometimes we don’t prepare the field on time because of which its cultivation gets delayed. We don’t get labor at the time of cultivation” (Male village head, age 55, Tati village, Torpa block).

Overall, participants reported that finger millet produced good size yields. However, they described the need for improvements in farming methods to further improve yields. For example, one participant described the need for it to “be grown in (a) scientific way and… proper way” (Female health and nutrition worker, age 45, Nichitpur village, Torpa block). One of the challenges identified related to its production was the dependence of the crop on weather. Strong sunlight was reported to cause the crop to die. In addition, heavy rain during production was described as turning the crop black, leading to it having a bitter taste.

Participants described two main benefits of finger millet cultivation. First, several participants described finger millet as easy to grow and requiring relatively low quantities of water. As one participant (Male village head, age 55, Charid village, Murhu block) stated: “Mandua farming is done easily. There is not much of hard work. Farming can be done with less water as well.” The second benefit was that production of finger millet supplemented rice production. One participant said, “Only rice is not enough. That’s why mandua farming should be done.” (Male village head, age 60, Nichitpur village, Torpa block).

### 3.2. Processing

Participants described the processing of finger millet as consisting of drying, cleaning and beating the crop. One participant summed up this process as follows: “Firstly, it is sun dried properly and then beaten up with a stick. Then it is cleaned and prepared for eating” (Female village woman, age 35, Nichitpur village, Torpa block). Another participant elaborated on the process of beating the millet, saying: “First we get it beaten in “thenki” (okhli), then get it grinded in the machine and then eat” (Male village head, age 47, Kudapurti village, Murhu block). Participants reported issues occurring during the processing such as hands swelling during the beating of the millet and the wind not blowing in the way needed to enable the cleaning of the crop. In addition, if rains occurred during harvesting “and if not dried properly then fungus appears” (Female health and nutrition worker, age 35, Charid village, Murhu block). A few participants noted using machines to grind finger millet for consumption.

### 3.3. Storage

Participants reported being able to store the dried and cleaned finger millet in sacks for a long period of time (e.g., up to a year). However, they reported experiencing postharvest losses due to insects, termites and rats eating the stored crop as well as spoilage from rain when the millet was improperly stored. While some participants described avoiding using insecticides or pesticides during storage, others reported using “powder to protect from termites” (Female health and nutrition worker, age 35, Charid village, Murhu block). One participant also mentioned the use of a tarpaulin to cover the crop during storage to prevent water damage. In terms of improving storage, one participant stated, “When we harvest the mandua crop, then there should be “machaan” (wooden structure made to store the harvested crops) in farm to keep the crop safe. Dried straws (puwaal) should be spread at the top of “machaan” (Female health and nutrition worker, age 35, Charid village, Murhu block). Two other participants suggested the need to check on the crop periodically during storage to ensure its safekeeping.

### 3.4. Retail

A few of the participants reported selling finger millet to a mahajan or baniya (tradesmen) in the market. As one participant described: “Mahajan or baniya” fixes the price and then they buy accordingly” (Female health and nutrition worker, age 35, Charid village, Murhu block). Several benefits of selling finger millet were identified by interviewees. These included being able to pay school fees of children or to enable the purchase of vegetables or other household items. One participant said: “Because of shortage of money and to fulfill our other needs, we sell mandua” (Male village head, age 55, Charid village, Murhu block). Participants indicated that selling finger millet was profitable given that it was sold at a higher price than rice. While interviewees noted benefits of selling finger millet, they also identified challenges. These included difficulties accessing the market, rate fluctuations and meeting demand. As one participant stated: “We have to go to market to sell mandua but (the) vehicle doesn’t reach to mahajan. In that case we have to carry and go there” (Female health and nutrition worker, age 35, Charid village, Murhu block). In addition, the lengthy cultivation process (4.5–5 months) creates a delay between the beginning of the cultivation process and receiving revenue from the sold crops.

### 3.5. Preparation & Consumption

Most participants said that they spent at least 30 min cooking finger millet. When asked about cooking fuel, participants stated that they typically use firewood given the high cost of gas. The participants mentioned several dishes that they made with the millet including: chilka chapatti or roti, “dombu (steamed finger millet balls)” and “halwa (sweet dish made by boiling finger millet flour with sugar and oil” or “kode lete/khichdi (dish made of finger millet, pulses and vegetables)”. Participants discussed safety concerns when asked about challenges during cooking. In particular, they mentioned the need to prevent their hands from burning. As one participant described: “In this, cooking should be done carefully, so that hands are not burnt while cooking” (Male village head, age 54, Tati village, Torpa block).

Interviewees identified several benefits related to consuming finger millet including its taste and health properties. More specifically, participants described it as tasting good and slightly sweet as well as containing vitamins and imparting strength. It was also described as an alternative to rice consumption. However, a couple of participants stated that they avoided eating the crop during the summer season because it made them feel thirsty.

### 3.6. Entry Points for Strengthening Supply Chain

Figure 4 provides an overview of the barriers and facilitators across the finger millet supply chain among the Munda tribe. There were several facilitators to its production but also barriers that could be addressed with interventions. One of the key entry points for strengthening the finger millet supply chain was to encourage interventions from agricultural extension organizations to ensure that best practices are used in agricultural production as well as postharvest processing, storage and sale. Agricultural extension should include access to improved (e.g., agronomic, nutritional) crop varieties and introduction to climate information such as drought management and improved planting schedules based on seasonal forecasts. Given the laborious nature of processing finger millet, increasing access to low-cost technology such as small machinery (e.g., destoner, grain polisher) that assists with milling would help reduce time and energy costs, particularly among women. In addition, farmers could be provided with market price information to better inform their decisions regarding the sale of millet. Increasing storage capacity and improving infrastructure, by managing pests and humidity, within villages would also help to reduce postharvest losses, thereby increasing the availability of finger millet. Lastly, increasing access to clean cooking fuel that is affordable is needed to help reduce safety concerns regarding millet preparation. This could also help to reduce exposure to air pollution related to cooking fuel; however, this was not a concern raised by participants.

## 4. Case Study 2: Koinaar Leaves Value Chain among the Sauria Paharia Tribe

### 4.1. Production/Harvest

Figure 5 provides an overview of the steps of the Koinaar leaves’ supply chain. Most participants stated that they collect Koinaar leaves in the forest or forest hills. The quantity of Koinaar leaves collected ranged from 1–3 kg per harvest, with the majority stating that they collect 1–2 kg. The consensus was that Koinaar leaves are collected between March and April (spring and summer seasons), while fewer participants stated that it was collected in March and August. Most participants described harvesting new leaves, which were described as soft and green, from the top of the branches: “We pluck soft leaves from the top side” (Female villager, age 27, Chota Haripur village, Sunderpahari block). Several participants noted methods for reaching the new growth such as climbing the tree, cutting branches, or cutting down the entire tree: “… when people are not able to climb on the tree… they cut the trees for leaves… and that’s why whole tree gets destroyed” (Female villager, age 27, Chota Haripur village, Sunderpahari block). Freshness was frequently cited as a desirable quality. A lesser number of participants referenced taste as a key characteristic when foraging for Koinaar leaves.

It was clear from the interviews that the community collected these leaves 2–3 times a week, when in season. The time spent in the collection of Koinaar leaves for one-time consumption was high. Traveling contributed to a major part of the long hours (1–5 h) spent on the collection of these leaves. Once in the forest, searching for the trees as well as searching for and collecting “soft” leaves was also a time-consuming process (0.5–4 h). Most participants reported spending one to two hours searching and collecting.

Participants were asked about their opinions on growing Koinaar leaves in their backyards and/or open fields. The majority (*n* = 13) felt that there was no reason to grow the tree, while a small number (*n* = 4) expressed that they would be interested in growing Koinaar leaves. One woman said: “Yes… it will be good… for good growth (of plant) we have to take care of it properly… then it will be good… we won’t need to go to forest anymore… we can have it nearby at any time… in forest monkeys used to eat all new leaves of Komo (Koinaar) tree… but if it will be in our fields then monkey will not eat and we will easily get fresh leaves” (Female villager, age 38, Chota Haripur village, Sunderpahari block).

Most of the participants expressed that they would need support in terms of external help, infrastructure and resources to grow Koinaar trees in the villages. One woman stated, “see when we will grow the trees the cattle will graze it all… so if we will get facility for fencing and irrigation… then it would be good…” (Female villager, age 32, Teletok village, Boarijor block). The need for additional external help was also described by the participants in the context of water for irrigation, organic fertilizers (i.e., cow dung), fencing (i.e., bamboo sticks) and tools (for digging). A few participants said that they would not need any external help as the “(tree) can grow on its own without fertilizers” (Elderly female villager, age 56, Chota Haripur Village, Sunderpahari block).

A few participants also highlighted that they themselves or their relatives including parents, grandparents and uncles had grown the plant in the past. One elderly woman said, “yes not recently… but my father grew it… near to road of that stall… it was present there…we could see it while grazing our cattle there…” (Elderly female villager, age 65, Chewo village, Sunderpahari block). However, these participants were in the minority because most participants had limited past experiences with the plant outside of foraging. The majority (*n* = 10) had never known the plant to be grown by family members.

### 4.2. Processing

One way to preserve Koinaar leaves is to dry them. However, when asked about the possibility of drying the leaves, many of the participants opined that drying would change the taste of the plant leaves and hence was not an option for preservation and storage. One young woman stated, “dry leaves doesn’t taste as good as fresh leaves…” (Female villager, age 21, Chewo village, Sunderpahari block). The majority simply stated that they had never tried drying the plant and had only consumed it fresh. Meanwhile, one woman believed that there was no reason to dry the plant because of its availability, stating “Komo Ghasi (Koinaar leaves) is available easily so don’t need to store it… like they never dry mustard leaves when in season…” (Female villager, age 38, Chota Haripur village, Sunderpahari block). However, a few agreed that drying Koinaar leaves could be a possibility: “we can dry Komo Ghasi (Koinaar leaves) and churn it into powder and then we can add that powder into pulses… that will be good… but people don’t use this… now it’s available in every alternate day (during season)…” (Elderly female villager, age 65, Chewo village, Sunderpahari block).

### 4.3. Retail

The selling of Koinaar leaves was not reported by the Sauria Paharia community. Several reasons were given which included: lack of additional collection of Koinaar leaves for sale as it is consumed a lot without any leftovers, no actual reason for selling, hesitation in the community due to perceived risk and belief that no one will buy it, and language barriers that might affect the interaction with the buyers. For example, one elderly woman stated that she believed the Paharia community did not sell because “hesitation and as we can’t speak Hindi…” (Elderly female villager, age 47, Teletok village, Boarijor block). However, the participants expressed that there could be a potential profit in selling Koinaar leaves. One woman said, “there is a profit because it doesn’t require any money in selling it” (Female villager, age 32, Kusumghati village, Boarijor block). Participants were also asked if they would buy Koinaar leaves if they were sold in the market. Most felt that there was no need to buy them because of their availability in the forest. However, one participant did feel that there was a possibility that people may buy Koinaar leaves in the future, while another participant stated that if they were no longer available in the forest (in the future) and were sold in the market then they would buy them. The respondents also shared that another Indigenous community ‘Santhals’ residing in the same geographical area, sell Koinaar leaves. They believed that the Santhals sell the plant while they don’t for several reasons including the Santhal community’s disinterest in consuming it, prior experience selling items in the market, their higher need for money, having less hesitation to sell and their ability to use their own language when selling.

### 4.4. Preparation & Consumption

Overall, there was a common consensus on cooking techniques for Koinaar leaves. Prior to cooking, some women reported sorting out the old leaves and stalks, removing the petioles from the new leaves and washing the leaves. One woman described the cooking process as follows: “After selecting soft leaves, we chop them in tiny pieces. And boil for some time… and after discarding that water just cook that with garlic, onion, chilli and salt.” (Female villager, age 35, Kusumghati village, Boarijor block). Several participants also noted that Koinaar leaves are primarily cooked on wood stoves in clay pots or a Degchi (aluminum pot). When discussing their past cooking experiences with the Koinaar leaves, a participant mentioned an ancient cooking method for the plant in a Daliya or a clay pot.

### 4.5. Entry Points for Strengthening Supply Chain

Figure 6 provides an overview of the entry points for strengthening the Koinaar leaves supply chain among the Sauria Paharia communities. One of the key barriers was the time it took women to travel to the forest and forage the new leaves. An additional barrier was the difficulty in accessing the new leaves on older trees resulting in individuals climbing or cutting down the trees. Given the challenges associated with acquiring the leaves from the forest, producing them within the villages could help improve access. However, there was little demand for this among the interviewees, which may be due in part to their limited experience growing it in the past. Increasing access to extension services focused on growing, drying and selling Koinaar leaves could help to overcome some of the barriers to its production, retail and consumption across seasons. Moreover, creating opportunities to sell directly to schools or other institutions, rather than in markets, may help to address some of the identified barriers to its sale.

## 5. Discussion

While the existing literature examines value chains of crops from India at an industrialized scale [13,14,15], research in wild edible and traditionally consumed staple crop value chains at a rural level is limited. Such research is critical to capture TEK for human and planetary health. Using value chain analysis, we examined the key barriers and facilitators to strengthening the value chains of two nutrient-rich plants among Indigenous populations in Jharkhand state, India: finger millet and Koinaar trees. Overall, we found both barriers and facilitators across the value chains of both foods. Increasing access to high quality agricultural extension would help to strengthen the production of both foods, while addressing barriers related to sub-optimal production practices (finger millet) and the significant amount of time and resources spent foraging (Koinaar leaves). For finger millet, improved agronomic practices and postharvest methods such as targeted use of fertilizer, improved crop varieties, machinery for processing, storage techniques and infrastructure and improved cooking fuel would help reduce barriers across the supply chain related to yield, postharvest losses, processing labor and safety concerns related to cooking. For Koinaar leaves the value chain could be strengthened through the improvement of drying techniques, focused on palatability and consumer acceptance, to increase access across seasons. In addition, providing retail training could increase opportunities for selling it outside of local markets, where participants mentioned potential language barriers. We provide additional insights into how to leverage both supply chains to improve diets and nutrition among Indigenous populations in Jharkhand state of India.

### 5.1. Production

Some ways to support the production of both nutrient-rich foods may involve providing agricultural extension services to improve production techniques for improved yields, incorporate use of technology (where appropriate) and increase market access. While Indigenous populations have a wealth of knowledge, this knowledge could be complemented with additional knowledge from agricultural extension services. However, there are limitations to the current extension services provided in India [26] and these services often do not reach underserved populations such as women and Indigenous populations [27,28]. The Munda tribe noted that access to extension services focused on improved cultivation techniques and varietal types could improve finger millet yield, increasing availability for both household consumption and sales [29]. A study by Pradhan and colleagues [30] found that use of an improved finger millet crop varietal (GPU-67) along with recommended cultivation techniques and nutrition education resulted in 60% higher grain yield, 1.16 times more profit, decreased women’s labor and an improvement in household consumption compared to baseline. Additional training focused on postharvest storage and retail sales could further improve all aspects of the finger millet value chain.

Interviewees among the Sauria Paharia community indicated that they spent a significant amount of time looking for and harvesting Koinaar leaves, including time spent traveling to the forest. The substantial amount of time spent foraging may signify an opportunity cost wherein community members are losing considerable time that might be used for other important activities. Moreover, since Koinaar trees are evergreen trees that reach approximately 15 to 20 feet in height, the new leaves, which are desirable for harvest, may continue to grow out of reach. As such, women noted having to climb or cut down the Koinaar trees to harvest the new leaves. One way to increase the availability of Koinaar leaves would be to increase their local production within the village. To support those who are interested in growing Koinaar leaves, external help and resources would be required to facilitate this production including provision of irrigation facilities, fertilizers, fencing and agronomic tools. Extension services could focus on production techniques such as the use of coppicing, a traditional tree management technique that promotes trees to grow new shoots from their felled stump. A study conducted by Singh and colleagues showed high yield of leaves from coppiced Koinaar trees [31]. This technique could promote the growth of new leaves, increase accessibility to leaves within reach and the production of wood for additional uses. Furthermore, this technique may decrease the number of trees being harvested in the forest, increasing crop sustainability.

Both communities expressed a demand for extension assistance to improve availability and accessibility of the nutrient-dense crop. Extension services in other settings have improved farmers’ livelihoods in several ways. The Krishi Vigyan Kendra (KVKs) (agricultural extension center in India) allows for education and innovative technology to be disseminated to farmers to improve food production. KVKs could be instrumental in supporting the Sauria Paharia community with the creation of community or home nutrition gardens to grow Koinaar trees and to support improved finger millet production practices among the Munda community. This could be a viable solution to increase availability of the leafy vegetable and eventually decrease the time spent in harvesting and foraging, freeing time for other activities among the Sauria Paharia as well as increasing finger millet yields among the Munda population. Increasing access to these nutrient-rich foods could help to increase micronutrient intakes among these communities, decreasing the burden of malnutrition [2,17].

### 5.2. Processing and Storage

While a substantial amount of the focus on strengthening value chains for improved nutrition is on improving production, strengthening processing and storage can also help to increase access to these foods, particularly across seasons. For both foods examined, we identified the need to improve processing as a key entry point for strengthening their supply chains. Given the limited seasonal availability of Koinaar leaves, drying the leaves could be one way to preserve them so they can be consumed year-round. Although drying would be an option, there was a lack of preference for this technique because of perceived changes in flavor. However, there are ways to optimize preservation to increase the nutrient and flavor retention [32,33]. In a review of postharvest preservation techniques used for African leafy vegetables, blanching, drying and fermentation were all described as suitable means for preservation [34]. Additional research and development by KVKs and nutrition professionals is necessary to develop drying methods that might minimize the loss of flavor, while retaining nutrients, which is important for consumption. Training in terms of how best to preserve the leafy greens, as well as how best to store them, should be combined with training in terms of cooking with the dehydrated greens to increase consumption of this leafy vegetable in the community across seasons.

A key barrier to the processing of finger millet was its labor-intensive nature. Adopting low-cost machinery and technology to help with the processing would increase efficiency and reduce time and energy spent during the processing step of the supply chain, increasing revenue. There has been an ongoing effort to alleviate the drudgery and other issues associated with processing finger millet, as well as other millets, with machinery in India and beyond [35,36,37,38]. One evaluation of an intervention conducted at six sites in India found that the introduction of small farm machinery led to a 35–90% reduction in time spent on processing [35]. Another case study found that the machinery in a Millet Processing Centre in Tamil Nadu, India had a 90% milling efficiency which increased overall revenue [37]. The input of similar small farm machinery by KVKs or NGOs might help the Munda community overcome this barrier during processing leading to enhanced value and improved processing time of finger millet. Efficient and effective processing can improve the crop’s storage capacity, which can be as long 10 years [26].

In addition to the need for the adoption of technology for processing finger millet, we identified the need for improved storage infrastructure and methods given the reported postharvest losses experienced due to pests and rain damage. A scoping review of interventions to address postharvest losses in sub-Saharan Africa and South Asia found storage interventions that focused on pesticide use, modified atmospheres, storage containers, or some combination of the three [39]. While some of these approaches may be feasible within the Munda community, others may not. However, there may be traditional indigenous storage methods that could be adopted to reduce postharvest losses. For example, storing grains in bamboo baskets lined with neem leaves or with leaves of other plants with preservation properties, and protected from the outside elements with dung, could be one of the solutions [40]. These traditional storage techniques have been used in other indigenous communities and may be feasible within the Munda tribe as well.

### 5.3. Retail

Among the Munda community, challenges related to the sale of finger millet included not having access to market price information. Such information could better inform farmers on when to sell. Furthermore, market information could potentially help farmers aggregate harvests for wholesale into larger markets. While some studies found positive impacts of providing smallholder farmers with market price information through mobile phones, the evidence is somewhat mixed [41]. Several challenges to selling Koinaar leaves were identified by the Sauria Paharia. These barriers related to the perceived lack of substantial profit due to its abundant availability in the forest, a lack of buyers and language barriers hindering their ability to communicate with consumers in the markets. Therefore, support may need to be provided to community members who wish to enter the markets. In place of selling in the markets, community members could sell directly to schools or Anganwadi centers (maternal and child health and nutrition center) for the government led supplementary feeding programs. An intervention creating a new avenue to sell Koinaar leaves may help overcome the language barrier as well as the lack of potential buyers. Value chain interventions have strengthened the retail sale of African leafy vegetables, despite their abundance in the natural food environment [42]. Such interventions have focused on retail best practices as well as improved infrastructure such as the development of wholesale markets, which allow farmers to aggregate their harvests and empower them to participate in larger market sales [43]. Lessons learned from the African leafy vegetable value chains could be used to support the finger millet and Koinaar leave value chains in India.

### 5.4. Preparation and Consumption

We found that one of the main challenges related to the preparation of finger millet was the potential for community members to burn their hands while preparing it given their reliance on wood fire stoves. Transitioning to improved cooking fuel could help to reduce some of the challenges associated with the preparation of finger millet. One study on Indigenous people in rural Nepal explored the implications of transitioning from the traditional cooking practice of using firewood to modern cooking solutions such as the use of biogas [44]. However, one of the barriers towards shifting to cleaner cooking fuel may be cost [45]. Subsidies for liquid petroleum gas (LPG) cooking fuel have played an important role in expanding access to cleaner cooking fuel in India, but the poorest and most vulnerable populations have not sufficiently benefited [46]. Ensuring that the LPG subsidies reach Indigenous populations in India could help to increase their access to clean and safer cooking fuel.

Overall, both finger millet and Koinaar leaves were considered desirable by participants. Interviewees described enjoying the taste of these foods as well as acknowledging their nutrition and health properties. Given the preference for both of these foods, alongside their nutrient-density, incorporating them into supplementary feeding programs (including school meals) may increase their consumption even further with the potential to help address malnutrition in these settings.

### 5.5. Limitations

One of the key limitations of this study was our sole reliance on qualitative data. Incorporating quantitative data into our analyses could have helped to triangulate our findings. In addition, future work could also include interviews with other value chain actors such as agronomists, KVKs, etc.

## 6. Conclusions

In this study, we conducted value chain analysis of finger millet with the Munda and Koinaar leaves within the Sauria Paharia communities to identify barriers and facilitators across their supply chains that influence their production, the way they move through the supply chain, and their consumption. Overall, improving extension services across the supply chains of both foods would help strengthen their production, storage, processing, retail and consumption. These extension services need to focus beyond production and integrate other steps of the supply chain to address the existing challenges faced by these communities such as improved storage facilities and postharvest handling methods. Given the high preferences for these foods, it is likely that addressing the current barriers in their supply chains would lead to significant improvements in production and consumption with the potential to help address the burden of malnutrition. Furthermore, these improvements could lead to increased surplus for retail improving the economic contribution of these indigenous foods to household income.

## Figures and Tables

**Figure 1 foods-11-03774-f001:**
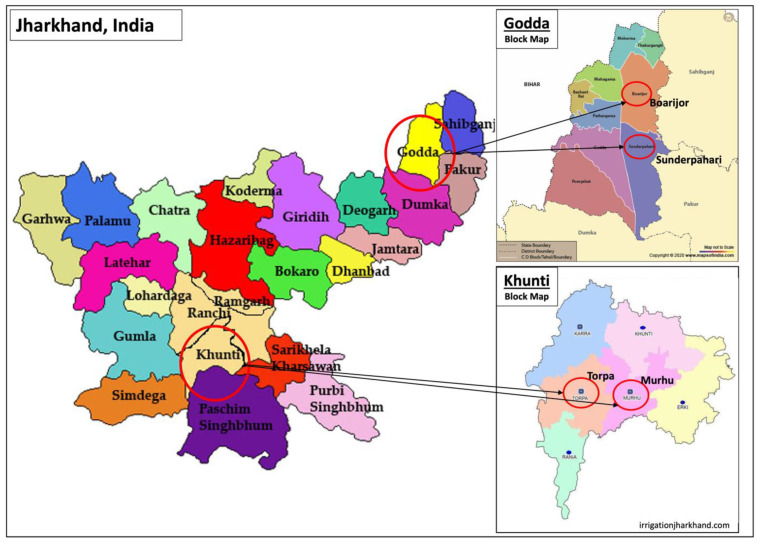
Selection of study blocks in Godda (Boarijor and Sunderpahari) and Khunti (Torpa and Murhu) districts of Jharkhand, India.

**Figure 2 foods-11-03774-f002:**
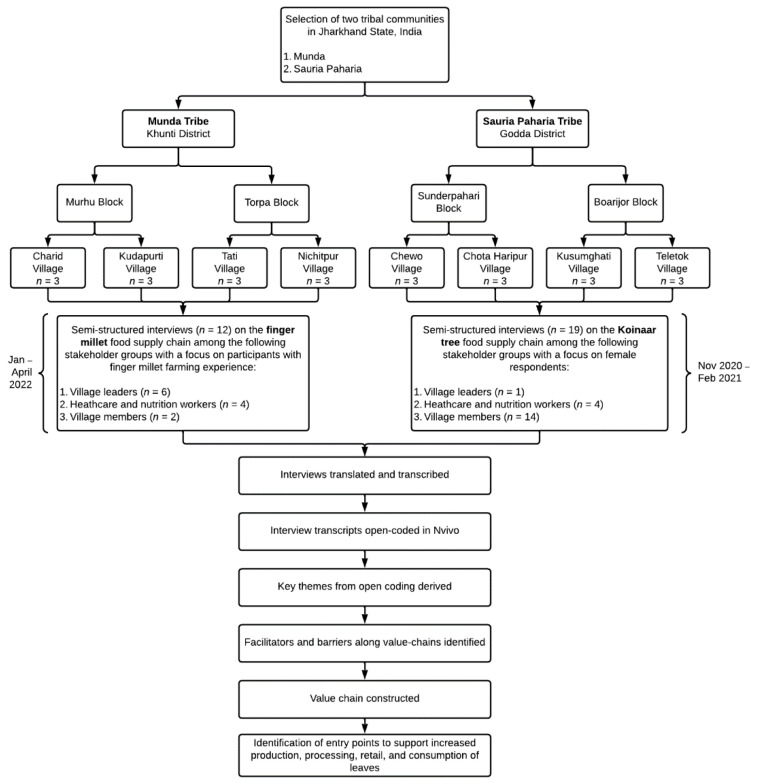
Schematic representation of study.

**Figure 3 foods-11-03774-f003:**
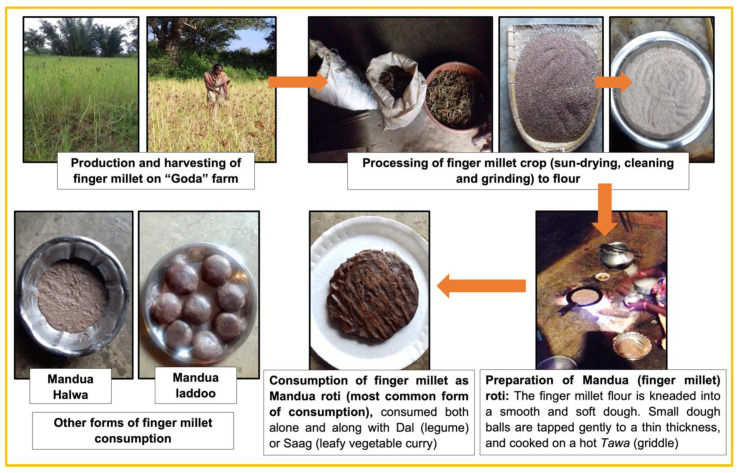
The different steps in the value chain of finger millet in Khunti district, Jharkhand, India.

**Figure 4 foods-11-03774-f004:**
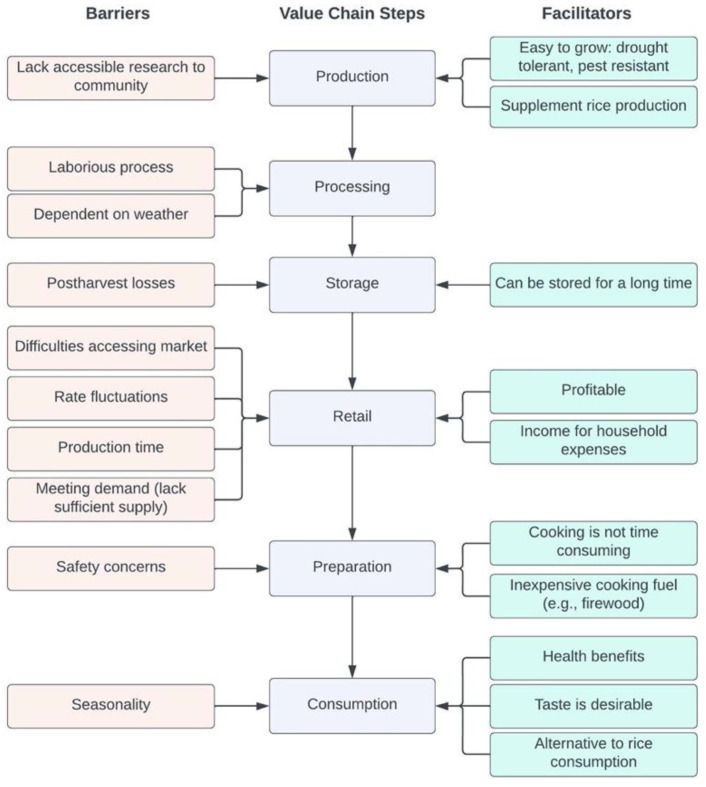
Overview of barriers and facilitators to the production and consumption of finger millet.

**Figure 5 foods-11-03774-f005:**
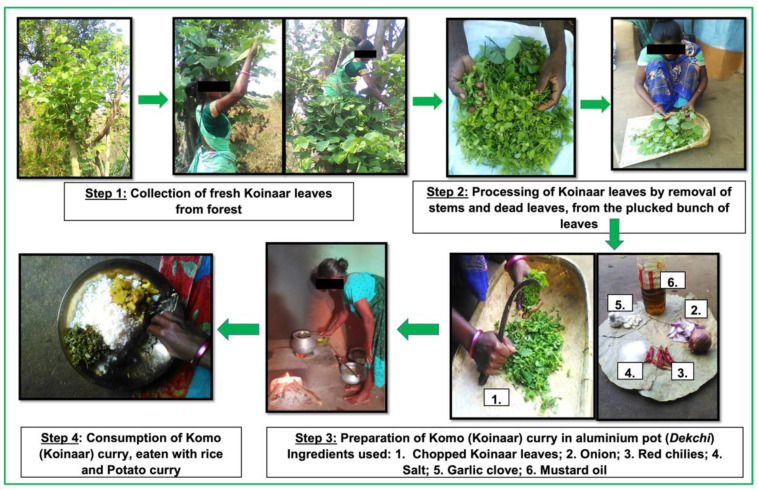
The different steps in the value chain of Koinaar leaves in Godda district, Jharkhand, India.

**Figure 6 foods-11-03774-f006:**
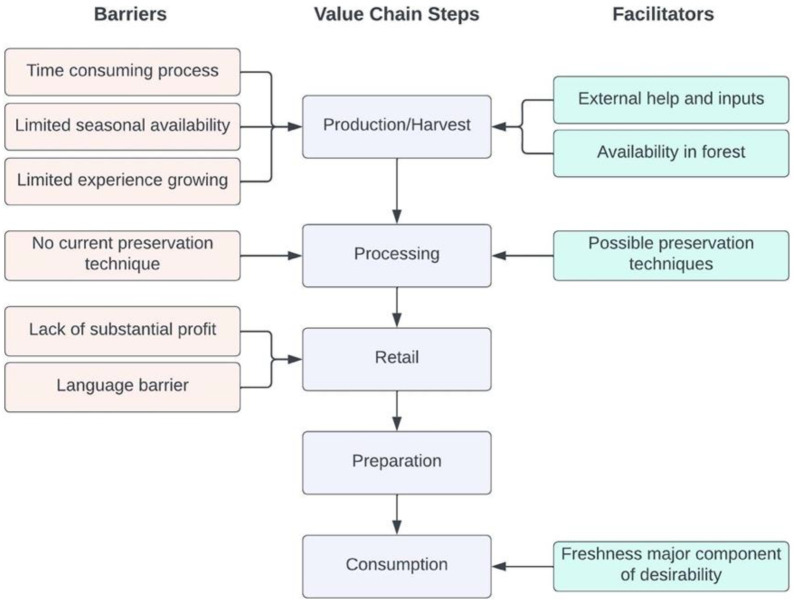
Overview of barriers and facilitators to the production and consumption of Koinaar leaves.

## Data Availability

The data presented in this study are available on request from the corresponding author.

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
