# Peer review of "Leveraging Nutrient-Rich Traditional Foods to Improve Diets among Indigenous Populations in India: Value Chain Analysis of Finger Millet and Kionaar Leaves"

_foods, 2022, doi:10.3390/foods11233774_

Round 1
Reviewer 1 Report
This manuscript describes two case studies on the potential of underutilised crops as sources of macro- and micronutrients among indigenous populations in India. The approach of the study is mainly based on quantitative tools (semi-structured interviews) for gathering information on production techniques, processes, storage conditions, retail options and conditions for preparation and consumption.
The manuscript is very well written, with a clear definition of the objectives and expected results. However, having a couple of schematic figures of the methodological approach will help readers, not so specialised in social sciences, to find the manuscript attractive.
Two schematic figures are suggested:
1) Methodological approach (including number of individuals "n" interviewed in each step), it is mentioned at some points, but is missing in others.
2) Flow diagram and figures also for case study 1 (finger millet).
Minor comment. In line 397, did the authors mean "An additional"? Please check.
Author Response
Reviewer 1 feedback:
This manuscript describes two case studies on the potential of underutilised crops as sources of macro- and micronutrients among indigenous populations in India. The approach of the study is mainly based on quantitative tools (semi-structured interviews) for gathering information on production techniques, processes, storage conditions, retail options and conditions for preparation and consumption.
The manuscript is very well written, with a clear definition of the objectives and expected results. However, having a couple of schematic figures of the methodological approach will help readers, not so specialised in social sciences, to find the manuscript attractive.
Two schematic figures are suggested:
1) Methodological approach (including number of individuals "n" interviewed in each step), it is mentioned at some points, but is missing in others.
Thank you for the kind recommendation. Please see the addition of a participant flow diagram (Figure 2).
2) Flow diagram and figures also for case study 1 (finger millet).
Thank you for the kind recommendation. Please see the addition of flow chart depicting the finger millet case study (Figure 3).
Minor comment. In line 397, did the authors mean "An additional"? Please check.
We have edited the manuscript to reflect the above comment.
Reviewer 2 Report
The work speaks about traditional food of the India region and in my opinion it has not ad international sounding, however it is not bad written and the main suggestions that I can do to authors are the following:
- - In the introduction section Authors should better stress why it would be relevant to develop a study of this type, what is the research gap to cover, and develop an explicit research question to be answered.
- I have never seen “Boxes” in a research paper, therefore I strongly suggest to move the content of BOX1in a paragraph of the paper, I suggest as part of the methodology section.
- - Authors stated “In previous work, we identified four of these tribes 111 to focus on, the Santhals, Ho, Munda, and Sauria Paharia”, but they need to precisely cite which previous work they refer to.
- - Limitations should be added to the conclusion section and not to the discussion one.
- - The discussion section missis the comparison with previous studies on this issue, and should be strengthened.
- - In the conclusion section it should be relevant to add more theoretical and practical implications deriving from the research study.
Author Response
Reviewer 2 feedback:
The work speaks about traditional food of the India region and in my opinion it has not ad international sounding, however it is not bad written and the main suggestions that I can do to authors are the following:
- In the introduction section Authors should better stress why it would be relevant to develop a study of this type, what is the research gap to cover, and develop an explicit research question to be answered.
To better stress why it is relevant to develop a study of this type and the research gap this manuscript covers, the following was added to the introduction:
“There is existing literature examining the value chains of crops in India such as pearl and finger millet [13-15]; however, this research tends to focus on longer, industrial value chains with policy recommendations targeting larger stakeholder groups and activities such as market aggregation and distribution. To capture traditional ecological knowledge for improved human and planetary health, it is critical that analyses of short value chains are conducted at a more local level for wild edible plants, such as the Koinaar tree, and traditionally consumed staple crops, such as finger millet. The current literature highlights the nutritional and ecological contributions of wild edible foods and traditional staple crops within Indigenous communities in India [2,3,5,6,8]; however, little is known about how their value chains can be leveraged to improve their production and consumption.
There is a clear need to conduct value chain analyses of underutilized crops within Indigenous communities in India given that it can help to identify how supply chains can be strengthened to improve their production, the way they move through the supply chain, as well as their consumption and contribution towards nutrient intake, thus adding nutritional value to the supply chain.”
In addition, we included the research question that this paper addresses: How can the value chains of the Koinaar tree and finger millet be leveraged to improve their production and consumption among Indigenous communities?
- I have never seen “Boxes” in a research paper, therefore I strongly suggest to move the content of BOX1in a paragraph of the paper, I suggest as part of the methodology section.
Thank you for the suggestion. The content of Box 1 was incorporated into the body of the methodology section: 2.2. Rationale for conducting value chain analysis of finger millet and Koinaar tree leaves.
- Authors stated “In previous work, we identified four of these tribes 111 to focus on, the Santhals, Ho, Munda, and Sauria Paharia”, but they need to precisely cite which previous work they refer to.
Thank you, the following citations were added to the statement.
- Ghosh-Jerath, S.; Kapoor, R.; Singh, A.; Downs, S.; Barman, S.; Fanzo, J. Leveraging Traditional ecological knowledge and access to nutrient-rich indigenous foods to help achieve SDG 2: an analysis of the indigenous foods of Sauria Paharias, a vulnerable tribal community in Jharkhand, India. Nutr. 2020, 7, 61, 1-26. doi:10.3389/fnut.2020.00061.
- Ghosh-Jerath, S.; Downs, S.; Singh, A.; Paramanik, S.; Goldberg, G.; Fanzo, J. Innovative matrix for applying a food systems approach for developing interventions to address nutrient deficiencies in indigenous communities in India: a study protocol. BMC Public Health 2019, 19, 944, 1-12, doi:10.1186/s12889-019-6963-2.
- Ghosh-Jerath, S.; Kapoor, R.; Barman, S.; Singh, G.; Singh, A.; Downs, S.; Fanzo, J. Traditional food environment and factors affecting indigenous food consumption in Munda Tribal community of Jharkhand, India. Nutr. 2021, 7, 1-18, doi:10.3389/fnut.2020.600470.
- Limitations should be added to the conclusion section and not to the discussion one.
Thank you for the suggestion. As per the Foods template the limitations are placed within the Discussion section.
- The discussion section missis the comparison with previous studies on this issue, and should be strengthened.
Thank you for this feedback. Based on our literature review, there is limited research on indigenous food value chains. While there is some research relative to crops in India in a global context, this research is novel in its approach to exploring an indigenous food system. To this effect, a paragraph was added to the introduction to provide context for this research paper (see comment above relative to the introduction). Furthermore, a summary of this paragraph was added to the first paragraph of the discussion as follows: “While the existing literature examines value chains of crops from India at an industrialized scale [13-15], research in wild edible and traditionally consumed staple crop value chains at a rural level is limited. Such research is critical to capture TEK for human and planetary health.”
The preceding paragraphs reference previous studies in comparison to our work where applicable. For example, line 614 “A study by Pradhan and colleagues…”
- In the conclusion section it should be relevant to add more theoretical and practical implications deriving from the research study.
To expand upon this section as suggested the following was added to the conclusion section:
“These extension services need to focus beyond production and integrate other steps of the supply chain to address the existing challenges faced by these communities such as improved storage facilities and postharvest handling methods. Given the high preferences for these foods, it is likely that addressing the current barriers in their supply chains would lead to significant improvements in production and consumption with the potential to help address the burden of malnutrition. Furthermore, these improvements could lead to increased surplus for retail improving the economic contribution of these indigenous foods to household income.“
Round 2
Reviewer 2 Report
Authors have now improved the paper.